# Thyroid hormone receptor α controls larval intestinal epithelial cell death by regulating the CDK1 pathway

Yuta Tanizaki[1], Hongen Zhang[2], Yuki Shibata [1] & Yun-Bo Shi [1✉]

Thyroid hormone (T3) regulates adult intestine development through T3 receptors (TRs). It is difficult to study TR function during postembryonic intestinal maturation in mammals due to maternal influence. We chose intestinal remodeling during *Xenopus tropicalis* metamorphosis as a model to study TR function in adult organ development. By using ChIP (chromatin immunoprecipitation)-Seq, we identified over 3000 TR-bound genes in the intestine of premetamorphic wild type or TRα (the major TR expressed during premetamorphosis)-knockout tadpoles. Surprisingly, cell cycle-related GO (gene ontology) terms and biological pathways were highly enriched among TR target genes even though the first major event during intestinal metamorphosis is larval epithelial cell death, and TRα knockout drastically reduced this enrichment. More importantly, treatment of tadpoles with cell cycle inhibitors blocked T3-induced intestinal remodeling, especially larval epithelial cell death, suggesting that TRα-dependent activation of cell cycle is important for T3-induced apoptosis during intestinal remodeling.

[1] Section on Molecular Morphogenesis, Cell Regulation and Development Affinity Group, Division of Molecular and Cellular Biology, Eunice Kennedy Shriver National Institute of Child Health and Human Development (NICHD), National Institutes of Health (NIH), Bethesda, MD, USA. [2] Bioinformatics and Scientific Programming Core, Eunice Kennedy Shriver National Institute of Child Health and Human Development (NICHD), National Institutes of Health (NIH), Bethesda, MD, USA. ✉email: Shi@helix.nih.gov

Thyroid hormone (T3) binds to T3 receptor (TR) and regulates developmental changes in many organs such as intestine and brain. In fact, hypothyroidism has been known to cause severe developmental defects in humans, including cretinism. TR has two subtypes, TRα and TRβ, in all vertebrates. The heterodimer formed between retinoid X receptor (RXR) and TR binds to T3 response elements (TREs) and recruits co-repressor complexes containing N-CoR or SMRT to suppress target gene expression in the absence of T3. After binding T3, TR-RXR heterodimer recruits co-activator complexes such as those containing P300 and SRC, and activates target gene expression[1]. T3 level peaks during mammalian postembryonic development, a period around birth when organ structure is changed from the fetal form to the adult form[2–4]. It is, however, difficult to study T3 function in this period in vivo by using mammalian models due to the maternal influence and difficult manipulation.

We have dedicated to study T3 function during vertebrate development by using the *Xenopus* model. *Xenopus* metamorphosis resembles postembryonic development in mammals and is totally dependent on T3. It is easy to manipulate amphibian metamorphosis and anurans like *Xenopus tropicalis* develop externally in a biphasic process that is independent of maternal influence. We have been studying intestinal remodeling to understand how T3 induces larval tissue degeneration and adult tissue development during metamorphosis. During intestinal metamorphosis, the larval epithelial cells undergo apoptosis while adult epithelial stem cells are formed de novo, followed by their proliferation and differentiation to form a complex, multiply folded adult epithelium[5–8]. In addition to epithelial transformation, the connective tissue and muscles also remodel extensively, accompanied by shortening of the intestine[5,9]. More recently, we and others have used genome-editing tools such as TALEN and CRISPR/Cas enzymes[10] to study the role of endogenous TR genes during metamorphosis[11–18]. These studies revealed an important role of endogenous TRs during intestinal remodeling.

Toward understanding the downstream pathways important for intestinal remodeling during metamorphosis, it is critical to identify TR target genes in the intestine. With the availability of an assemble *Xenopus tropicalis* genome and homozygous TRα knockout (TRα$^{-/-}$) tadpoles, we have carried out ChIP-Seq analysis for TR binding in the intestine of premetamorphic wild-type and TRα$^{-/-}$ tadpoles with or without 18 h T3 treatment. We observed that most of the TR-binding sites were bound by TR in the wild-type and TRα$^{-/-}$ tadpoles independent of T3 treatment, suggesting that TR binds to the targets constitutively and that most of the targets are bound by TRβ in the TRα$^{-/-}$ tadpoles. Bioinformatics analyses of TR bound genes and comparison with published RNA-Seq data[19] revealed numerous T3-induced programs, including cell cycle/proliferation pathways. Furthermore, we observed that treatment of tadpoles with cell cycle inhibitors blocked T3-induced larval epithelial cell death and intestinal shortening. Our findings suggest that T3 induction of cell cycle pathway is an important early event in cell fate determination during intestinal remodeling.

## Results

### ChIP-Seq identification of candidate TR target genes during intestinal remodeling.
We and others have recently shown that knocking out of either TRα or TRβ delays intestinal remodeling during natural and/or T3 induced metamorphosis[11,13–15,18]. RNA-Seq analysis of the intestine of premetamorphic wild-type and TRα$^{-/-}$ tadpoles treated with or without T3 for 18 h revealed that most of the T3 regulated genes were dependent on TRα, suggesting that TRα plays a dominant role early during T3-induced intestinal remodeling[19]. As these T3-regulated genes can

be either directly or indirectly regulated by T3, we decided to carry out ChIP-Seq analysis to identify direct TR target genes. This is because such genes are more likely to play critical roles in the earliest stages of cell fate determination during intestinal remodeling, a process involving both specific apoptotic degeneration of the larval epithelium and de novo formation of adult stem cells through dedifferentiation of some larval epithelial cells[8]. To determine if such TR target genes are TRα-specific, we carried out ChIP-Seq analysis, by using a polyclonal antibody that recognizes both TRα and TRβ in *Xenopus tropicalis*[20], on the intestine from both wild-type and TRα knockout tadpoles at stage 54 with or without 18 h of 10 nM T3 treatment as we previously used for RNA-Seq analysis[19].

To ensure the success of the ChIP-Seq analysis with the polyclonal TR antibody (Supplementary Fig. 1a), we first validated the ChIP samples after purification of ChIP DNA, prior to sequencing, by analyzing TR binding to the well-known TRE in the TRβ gene. As expected, TR was bound to the TRE and T3 treatment led to increased binding in wild-type intestine. On the other hand, this binding was reduced in TRα$^{-/-}$ intestine. Furthermore, when we analyzed TRβ gene exon 5, which has no known TR-binding site, we observed no TR binding in any samples (Supplementary Fig. 1b), demonstrating the success of the ChIP with the TR antibody. We next carried out sequencing on three technical replicates of these ChIP DNA samples on Illumina HiSeq2500 (100 bp read length). The raw paired-end sequencing reads from each sample were aligned to Ensemble genome assembly and xenbase v9.0 *Xenopus tropicalis* genome assembly (Ensembl; 19921, Xenbase; 34229) for genome-wide identification of binding sites (peaks) based on fragment pileup values at each nucleotide position. The ChIP-Seq FASTQ sequences were aligned to the *Xenbase* genome sequence to identify the pileup peaks and determine the peak heights. The peak information for different samples is shown in Supplementary Fig. 2. Most of peak widths were less than 500 bp and the number of peaks were more 20,000 (Supplementary Fig. 2) for each replicate of different samples.

As there would be expectedly some variations among the replicates that might produce false positive peaks, the peaks that were present in all three replicates would be much more likely to be true TR-binding sites. Thus, to focus on such peaks, we compared the peaks in the three replicates of each sample to identify peaks present in the same locations in all three replicates. This analysis yielded 11,469 and 17,930 peaks (Supplementary Fig. 2), corresponding to 3308 and 4319 TR-bound genes, for wild-type intestine without and with T3 treatment, respectively, or a total of 4605 TR bound genes (Fig. 1a, Supplementary Data 1–3), although most of peaks were detected in the intergenic region (Supplementary Data 4). Among the TR-bound genes, 3022 genes were common, indicating most genes were bound by TR constitutively (Fig. 1a and Supplementary Data 3). These TR-bound genes for the wild-type intestine and similarly identified TR-bound genes for the TRα$^{-/-}$ intestine (see below) were used for the rest of the analyses.

It is well known that the consensus TRE is the one made of two direct repeats of AGGTCA separated by 4 base-pairs (the idealized DR4 TRE). Consistently, we found more than 10,000 DR4 type of TREs in the regions bound by TR based on Gagne R et al.[21]. Importantly, when HOMER software was used to search for motifs in the sequences of all peaks from ChIP-Seq in the intestine of wild-type animals with or without T3 treatment, we found that binding sites for TRα and TRβ (i.e., THRa and THRb in Supplementary Fig. 3), and DR4 TRE sequences (i.e., LXRE Supplementary Fig. 3) were highly enriched among the TR-binding peaks, revealing a the consensus TRE sequence of GGTCA and (A/G)GGTCA separated by 4 bases, basically

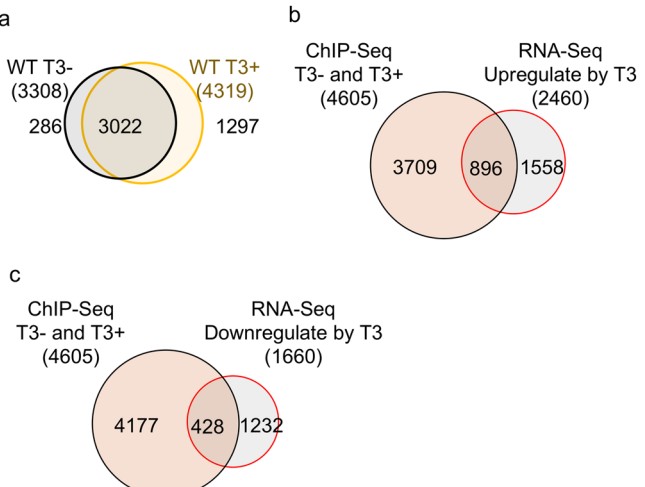

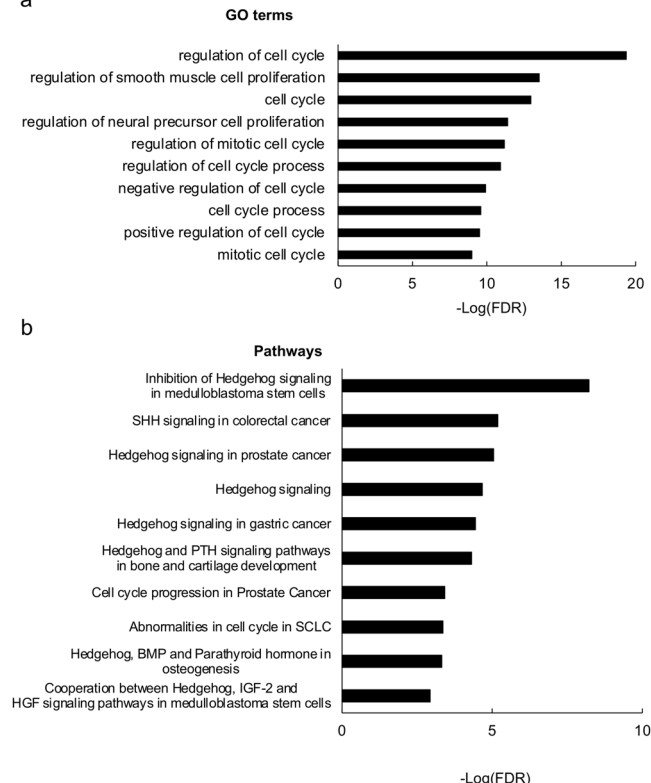

**Fig. 1 Most TR-bound genes found by ChIP-Seq are bound by TR constitutively in the wild-type intestine and many are upregulated by T3. a** Venn diagram of the TR target genes identified by ChIP-Seq in wild-type intestine with or without T3 treatment. The 3308 and 4319 TR-bound genes were identified from the 11,469 and 17,930 ChIP-seq peaks present in all three replicates of the −T3 and +T3 samples, respectively, based on the presence of TR-binding peak(s) in the coding region (including the introns) and/or 5 kb upstream or downstream of the cording region. Those peaks that could not be assigned to any known genes are indicated as "NA or Not Annotated" in Supplementary Data 1. Note that most genes were bound by TR constitutively. **b** Venn diagram showing overlap between the TR-bound genes identified by ChIP-Seq and T3-upregulated genes in the intestine based on previous RNA-seq study except using Xenbase for gene annotation and a cutoff of 1.5 fold regulation (cut-off threshold, FDR < 0.05)[19]. Of the 4605 TR-bound genes identified by ChIP-Seq, 19% were shown as the T3-upregulated genes. Conversely, 36% of the T3-upreguated genes were found to be TR-bound. **c** Venn diagram showing overlap between the TR-bound genes identified by ChIP-Seq and T3-downregulated genes in the intestine based on previous RNA-seq study except using Xenbase for gene annotation and a cutoff of 1.5 fold regulation (cut-off threshold, FDR < 0.05)[19]. Of the 4605 TR-bound genes identified by ChIP-Seq, 9% were shown as the T3-downregulated genes. Conversely, 26% of the T3-downreguated genes were found to be TR-bound.

**Fig. 2 GO terms and biological pathways related to cell cycle or hedgehog signaling that are enriched among TR-bound genes in intestine. a** Top 10 enriched GO terms related to cell cycle. The GO terms that were enriched among genes bound by TR in the wild-type intestine were ranked based on FDR as shown in Supplementary Data 5. The most significant GO terms related to cell cycle were plotted here (see Supplementary Data 18 for raw data). **b** Top 10 enriched pathways related to cell cycle or hedgehog signaling. The biological pathways that were enriched among genes bound by TR in the wild-type intestine were ranked based on FDR as shown in Supplementary Data 6. The most significant pathways related to cell cycle or hedgehog signaling were plotted here (see Supplementary Data 19 for raw data).

identical to the idealized DR4 TRE. Among them included previously reported TREs in *Xenopus* T3-regulated genes such as TRβ and TH/bZip genes[20] (Supplementary Fig. 4). In addition, we carried out independent ChIP-PCR assay to confirm TR binding to the TR-binding genes identified by ChIP-Seq analysis (see muc3a and atp6v1b2 as two examples in Supplementary Fig. 5).

To determine if the TR-bound genes were regulated by T3, we compared the TR-bound genes with genes upregulated in the wild-type intestine after 18 h T3 treatment as identified by RNA-Seq analysis[19]. We found that 19% of the TR-bound genes were upregulated by T3 while 36% of T3 upregulated genes were bound by TR (Fig. 1b). Supplementary Fig. 4 shows two examples of such upregulated genes, TRβ and TH/bZip, whose visualized ChIP-Seq and RNA-Seq mapping data suggest that increased RNA expression depends on TR-binding. In addition, 9% of the TR-bound genes were downregulated by T3 while 26% of T3 downregulated genes were bound by TR (Fig. 1c). Such percentages of overlaps between T3-upregulated genes identified by RNA-Seq and TR-bound genes identified by ChIP-Seq are highly significant, especially considering at least factors that reduce such overlaps. First, the short time of the T3 treatment and only a single time point used in RNA-Seq analysis would

miss many genes regulated by T3 at different time points; second, some T3-regulated genes from RNA-Seq analyses are likely regulated by T3 indirectly and thus do not have TREs.

**Gene networks directly regulated by TR in wild-type intestine.** To determine the biological processes and signaling pathways directly regulated by TR, we carried out bioinformatics analyses on the 4605 genes bound by TR in the wild-type intestine. GO analysis revealed that, not surprisingly, development-related GO terms were highly enriched among the genes bound by TR (Supplementary Data 5). In addition, many apoptosis-related GO terms were also found to be enriched among the TR-bound genes (Supplementary Data 5), in agreement with the fact that the first major change to take place during intestinal metamorphosis is the degeneration of larval epithelium via programmed cell death, which takes place before massive proliferation of adult epithelial stem cells[5,22]. Interesting, we found that many cell cycle/pro-liferation-related GO terms were highly enriched among the TR-bound genes (Fig. 2a and Supplementary Data 5). This finding raises a surprising possibility that cell cycle activation may be an important early stem for larval epithelial cell death during intestinal metamorphosis. Consistent with this, the pathway analysis showed that among the TR-bound genes, cell cycle/

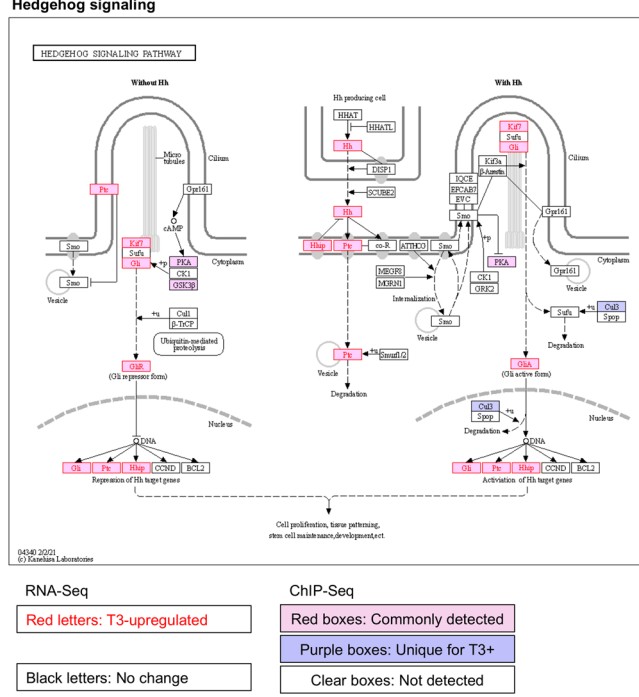

**Fig. 3 TR-binding and regulation of genes in the hedgehog signaling pathway in intestine based on ChIP-Seq and RNA-Seq data.** The hedgehog signaling pathway was visualized with regard to genes regulated by T3 based on RNA-Seq and/or bound by TR based on ChIP-Seq. The arrows show functional interaction: green for activation; red for inhibition. Letters in red indicate upregulated genes in wild-type intestine after T3 treatment. Letters in black indicate genes with no change in wild-type intestine after T3 treatment. Red boxes indicate ChIP-Seq detected genes both with and without T3 treatment, and purple boxes indicate genes identified by ChIP-Seq only in wild type with T3 treatment. Note that many genes including Sonic hedgehog (Hh) were upregulated by T3 and also bound by TR. No downregulated genes were found by RNA-Seq in this pathway.

proliferation pathways were enriched along with signaling pathways known to be important for apoptosis and for stem cells, such as Sonic hedgehog, Notch, and WNT/β-catenin pathways[23], (Fig. 2b and Supplementary Data 6). For example, the hedgehog signaling, as visualized along with T3-regulated expression data obtained from RNA-Seq[19], had many genes in the early phase of the signaling process, including SHH, CDON, HIP and PTCH1, bound constitutively by TR and upregulated by T3 (Fig. 3). Thus, the stem cell-related signaling processes are likely induced early by TR to facilitate the dedifferentiation of some larval epithelial cells to become adult stem cells[5–8]. The cell cycle program may be regulated early by TR to facility larval epithelial cell death since it is the first major event to take place, although it is also possible to be involved in cell proliferation following the formation of the stem cells.

**TRα knockout reduces the number of genes bound by TR in the intestine**. TRβ expression is low in premetamorphic intestine but is strongly upregulated at the climax of metamorphosis[20]. In contrast, TRα mRNA level is high in premetamorphic intestine and changes little during intestinal metamorphosis. Thus, we reasoned that TRα might play a dominant role in pre-metamorphic tadpole intestine. To investigate this possibility, we carried out ChIP-Seq analysis on the intestine from pre-metamorphic TRα$^{-/-}$ tadpoles treated with or without T3 as above and identified 2892 and 2469 genes bound by TRβ in the

TRα knockout intestine (as TRβ was the only TR expressed in TRα knockout animals) in the absence or presence of T3, respectively (Supplementary Data 7–9). Venn diagram analyses revealed that most of genes bound by TRβ in TRα$^{-/-}$ animals were independent of T3 treatment, i.e., common between TRα$^{-/-}$ tadpoles treated with T3 and without T3 (Fig. 4a, Supplementary Data 9, 10), again indicative of constitutive binding by TR as observed for the wild-type animals. In total, there were 3327 genes bound by TRβ in the TRα$^{-/-}$ intestine, of which only 7% were upregulated by 18 h T3 treatment in TRα$^{-/-}$ intestine based on RNA-Seq analysis (Fig. 4b)[19]. Of the 852 T3-upregulated genes in intestine of TRα$^{-/-}$ tadpoles, 27% were bound by TRβ in the TRα$^{-/-}$ intestine. Thus, the common genes between TR-bound and T3-upregulated genes were lower in TRα$^{-/-}$ intestine than those in wild-type intestine (Fig. 1b), suggesting that TRα play an important role in gene activation during the 18 h T3 treatment.

When we compared the ChIP-Seq data between wild-type and TRα$^{-/-}$ intestine, we found that of the 4605 TR target genes in wild-type intestine, nearly 70% or 3129 genes were bound by TR in TRα$^{-/-}$ intestine, while 94% of 3327 genes bound by TR in TRα$^{-/-}$ intestine were bound by TR in wild type (Fig. 4c, Supplementary Data 10), demonstrating high producibility of the our ChIP-Seq assay and suggesting that both TRα and TRβ can bind to the same target genes. Importantly, when we compared the 1476 genes bound by TR only wild type or 3129 genes bound by TR in both wild-type and TRα$^{-/-}$ intestine with the 896 genes bound by TR and upregulated by T3 (Fig. 1b), we found that about 19% of the TR-bound genes in either case were upregulated by T3 (Fig. 4d, e), much higher than the 7% of the TR-bound genes in TRα$^{-/-}$ intestine that were upregulated by T3 treatment of TRα knockout tadpoles. Furthermore, a heat map showing T3-induced fold change in gene expression in wild-type and TRα$^{-/-}$ intestine revealed that much higher fractions of the TR-bound genes were upregulated or downregulated by T3 for the wild-type-specific or common TR-bound genes than TRα$^{-/-}$-specific TR-bound genes (Fig. 5). In addition, among the genes regulated by T3, TRα knockout reduced T3-regulation, i.e., lower folds of upregulation or downregulation (Fig. 5). Thus, the genes bound by TR only in the wild-type intestine or in both wild-type and TRα$^{-/-}$ intestine respond to T3 similarly in the wild-type animals. More importantly, the findings indicate that TRα$^{-/-}$ is critical for the regulation of the genes by T3.

**Identification of GO terms and pathways enriched in a TRα-dependent manner among TR-bound genes**. By comparing the GO terms and pathways enriched among TR-bound genes in the wild-type intestine (Supplementary Data 5 and 6, respectively) and TRα$^{-/-}$ intestine (Supplementary Data 11 and 12, respectively), we obtained GO terms (Supplementary Data 13) and pathways (Supplementary Data 14) that were enriched among TR-bound genes in wild-type intestine but not enriched among TR-bound genes in the TRα$^{-/-}$ intestine. These GO terms and pathways are likely important for the observed effects on intestinal metamorphosis due TRα knockout. We found that many of these TRα-dependent GO terms and pathways were related to apoptosis. Their lack of enrichment may be responsible for the drastic inhibition of intestinal larval epithelial cell death within the first 3 days of T3-induced metamorphosis in TRα$^{-/-}$ tadpoles[19]. In addition, there were also many TRα-dependent GO terms (Supplementary Data 13) and pathways (Supplementary Data 14) related to stem cells, which may explain the lack of adult intestinal epithelial cell proliferation within the first 3 days of T3-induced metamorphosis in TRα$^{-/-}$ tadpoles[19].

Importantly, as with all enriched GO terms or pathways in the wild-type intestine, we found many cell cycle/proliferation-related

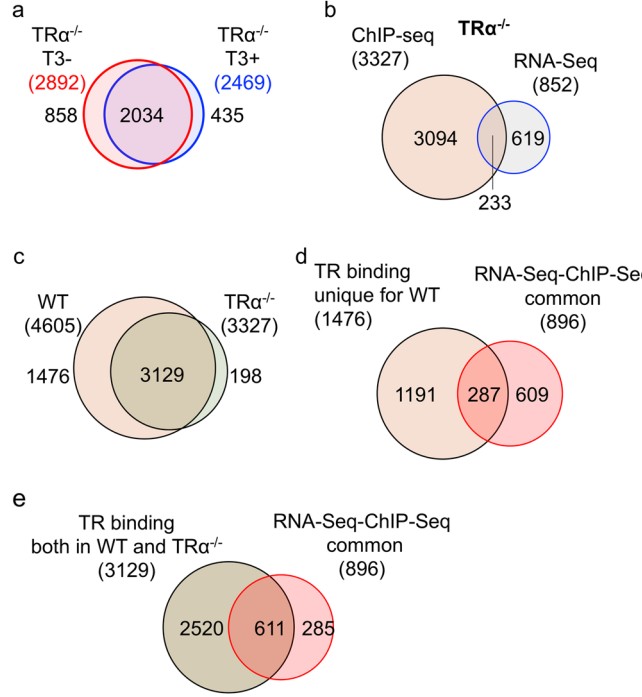

**Fig. 4 Most of TR-bound genes are common between wild-type and TRα−/− intestine with or without T3 treatment. a** Venn diagram of the TR target genes identified by ChIP-Seq in TRα−/− intestine with or without T3 treatment. Note that like in the wild-type intestine (Fig. 1a), most genes were bound by TR constitutively. **b** Venn diagram showing overlap between the TR-bound genes identified by ChIP-Seq in TRα−/− intestine and T3-upregulated genes in the intestine based on previous RNA-seq study[19]. Of the 3327 TR target genes identified by ChIP-Seq, 7% were shown as the T3-upregulated genes in TRα−/− intestine. Conversely, 27% of the T3-upreguated genes in TRα−/− intestine were found to be TR-bound. Both ratios were lower than those for the wild-type animals (Fig. 1b). **c** Venn diagram of comparison of genes detected by ChIP-Seq between wild-type (WT) and TRα−/− intestine. Of the 4605 TR target genes in wild-type intestine, nearly 70% or 3129 genes were bound by TR in TRα−/− intestine, presumably by TRβ, indicating that most of TR target genes in the intestine are commonly recognized by both TRα and TRβ. **d** Venn diagram comparison was made between 896 genes upregulated by T3 among all TR-bound detected in the wild-type intestine (Fig. 1b) and 1476 genes bound by TR only in the wild-type intestine (**c**). Of the 896 T3-upregulated genes that are bound by TR in wild-type intestine, 32% genes were genes bound by TR only in wild-type intestine, similar to the percentage of the genes bound by TR only in the wild-type intestine within all genes bound by TR in the wild-type intestine (1476 out of 4605). Similarly, of the genes bound by TR only in the wild-type intestine, 19% were upregulated by T3, just like the fraction of all TR-bound genes upregulated by T3 (Fig. 1b). **e** Venn diagram comparison was made between 896 genes upregulated by T3 among all TR-bound detected in the wild-type intestine (Fig. 1b) and 3129 genes bound by TR in both wild-type and TRα−/− intestine (**c**). Of the 896 T3-upregulated genes that are bound by TR in wild-type intestine, 68% genes were genes bound by TR in both wild-type and TRα−/− intestine, similar to the percentage of the genes bound by TR in both wild-type and TRα−/− intestine within all genes bound by TR in the wild-type intestine (3129 out of 4605). Similarly, of the genes bound by TR in both wild-type and TRα−/− intestine, 19% were upregulated by T3, again just like the fraction of all TR-bound genes upregulated by T3 (Fig. 1b). The results in D/E suggest that the genes bound by TR only in the wild-type intestine or both wild-type and TRα−/− intestine respond to T3 similarly in the wild-type animals.

GO terms (Supplementary Data 13) or pathways (Fig. 6a and Supplementary Data 14) that were enriched in the wild-type intestine but not TRα knockout intestine. For example, many genes in the pathway "chromosome condensation in prometaphase", including Cyclin A and CDK1, were bound by TR in the wild-type tadpole intestine but only two remained bound by TR in the TRα−/− intestine (Fig. 6b). Furthermore, RNA-Seq analysis showed that most of the TR-bound genes in this pathway were upregulated after 18 h T3-treatment (Fig. 6b and Supplementary Data 15)[19]. There were also several genes in this pathway that were not bound by TR but upregulated by T3 treatment (Fig. 5B), most likely indirectly. On the other hand, only a single gene, not bound by TR and thus likely indirectly, was downregulated by the T3 treatment (Fig. 6b). These findings suggest that such cell cycle/proliferation pathways are activated early during T3-induced intestinal metamorphosis.

**T3-induced activation of cell cycle program is important for larval epithelial cell death during intestinal metamorphosis.** To investigate the potentially surprising role of cell cycle program during intestinal metamorphosis, we focused on signaling pathways involving CDK1 by using a well-characterized inhibitor for CKD1, JNJ-7706621, to treat tadpoles in the presence of T3 (Fig. 7a). Thus, premetamorphic wild-type tadpoles at stage 54 were treated with or without T3 to induced precocious metamorphosis in the presence or absence of JNJ-7706621. We found that tadpoles treated with both T3 and JNJ-7706621 died after 3 days of treatment and therefore analyzed the animals after 2 days of treatment. As expected, after 2 days of T3 treatment, intestinal metamorphosis was initiated as reflected by the reduction in the length of the intestine (Fig. 7b). Furthermore, staining the intestinal cross-sections with methyl green pyronin Y for tissue morphology and TUNEL labeling for apoptosis revealed T3-induced epithelial folding and larval epithelial cell death (Fig. 7c, d) after 2 days of T3 treatment. Importantly, all these T3-induced changes were inhibited by JNJ-7706621, indicating that CDK1-dependent cell cycle program is important for T3-induced larval epithelial cell death.

To further test the role of the CDK1-dependent cell cycle pathway, we analyzed the role of Forkhead Box O1 (FOXO1), which has been shown to have a direct link downstream of CDK1 and is involved in cell death, repair of DNA damage and tumor suppression[24,25]. Thus, we also used a FOXO1 inhibitor, as1842856[26,27] to investigate if FOXO1 is required for T3-induced intestinal remodeling. The results showed that like JNJ-7706621, as1842856 treatment also led to animal death after 3 days and more importantly, inhibited all T3-induced changes in the intestine after 2 days of treatment. Thus, T3-induced activation of cell cycle program is an important early step in intestinal metamorphosis.

## Discussion
Molecular and genetic studies, especially recent analyses with knockout animals, have demonstrated important roles of TRs in mediating the effects of T3 during *Xenopus* metamorphosis, a model for understanding the mechanisms of mammalian post-embryonic development. The identification of TR target genes is thus critically important. Our studies here represent the first ChIP-Seq analysis for TR target genes during amphibian metamorphosis and demonstrate that many GO terms and biological pathways enriched among TR-bound genes are consistent with larval epithelial cell death and adult stem cell development and proliferation as the major cellular changes during intestinal remodeling. More importantly and surprisingly, we have found

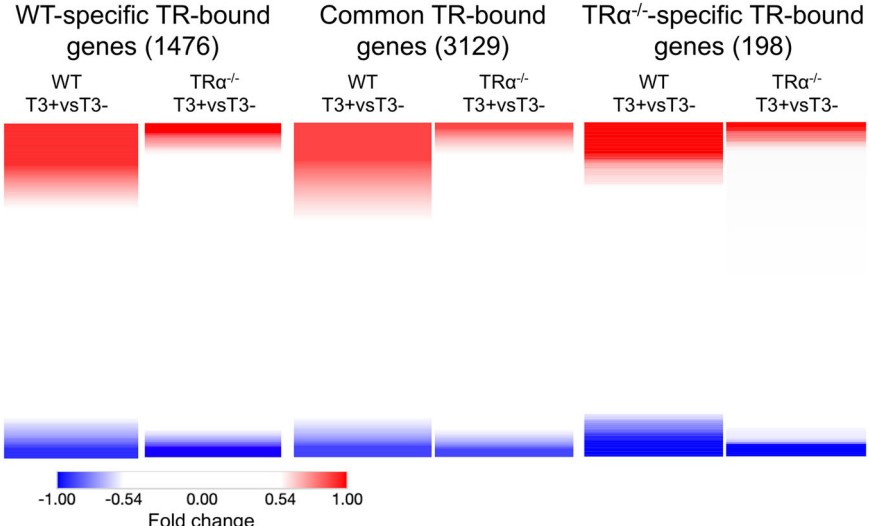

**Fig. 5 T3 regulation profile of ChIP-Seq detected genes.** Heatmap showing the fold change of gene expression in wild-type or TRα$^{-/-}$ intestine after 18 h T3 treatment as measured by RNA-Seq[19], for TR-bound genes detected by ChIP-Seq in only the wild-type intestine (left), both wild-type and TRα$^{-/-}$ intestine (middle) or only TRα$^{-/-}$ intestine (right). Note that higher fractions of the TR-bound genes were upregulated (red) or downregulated (blue) by T3 were present in the wild-type-specific or common TR-bound genes than TRα$^{-/-}$-specific TR-bound genes and that TRα knockout reduced T3-regulation, i.e., leading to lighter red or blue, for individual genes, suggesting that TRα$^{-/-}$ is important for gene regulation during the treatment. Note that the blank region between the red and blue areas are genes whose expression has no or little change after T3 treatment of wild-type or TRα knockout animals. The color range shows fold changes in log2 scale.

the GO terms and biological pathways related to cell cycle/proliferation are also enriched among the TR-bound genes, suggesting an important early role of cell cycle activation in T3-induced larval epithelial cell death, the first major cellular changes during intestinal metamorphosis. Consistently, we have demonstrated experimentally that inhibiting cell cycle blocks T3-induced intestinal epithelial cell death and morphological changes.

Our ChIP-Seq analyses showed that interestingly, most of the genes detected by ChIP-Seq with a polyclonal TR-antibody recognizing both TRα and TRβ were common in the wild-type and TRα$^{-/-}$ intestine with or without T3 treatment (Figs. 1 and 4, Supplementary Data 10). These results support the idea that TRs constitutively bind to TREs in chromatin and that most TR target genes are bound by both TRα and TRβ (as only TRβ is present in TRα$^{-/-}$ intestine)[28]. On the other hand, there were more genes bound by TR in T3-treated wild-type intestine than the intestine without T3 treatment, consistent with increased TR binding to target genes in the presence of T3 (e.g., Supplementary Fig. 1)[29] and thus genes with lower affinity TREs could be detected in the presence of T3 but not in the absence of T3. In addition, it is known that TR binding to some known TR target genes is reduced in TRα$^{-/-}$ animals compared to wild-type ones (e.g., Supplementary Fig. 1)[18], and ligand binding may stabilize the DNA binding by TR-RXR[30]. Since it is impossible for quantitative comparison between wild-type and TRα$^{-/-}$ intestine by using our current ChIP-Seq protocol, it is possible that some TR target genes common between the wild-type and TRα$^{-/-}$ intestine may have lower levels of TR binding in the TRα$^{-/-}$ intestine.

Comparison of our current ChIP-Seq data with the earlier RNA-Seq data[19] showed that 19% of the TR-bound genes were upregulated by T3 in the intestine within 18 h of T3 treatment, the same condition used for our ChIP-Seq analysis. In addition, 9% of the TR-bound genes were downregulated by T3. It is worth noting that some TR-bound genes may require longer T3 treatment time, be regulated by T3 in a tissue-specific manner, or have relatively small change in gene expression. Thus, the observation that 28% (19% upregulated plus 9% downregulated) of the TR-

bound genes was upregulated by T3 treatment in the intestine provides independent support that the TR-bound genes found here are true TR target genes. Conversely, we also found that 36% of T3-upregulated genes discovered by RNA-Seq analysis were among the genes bound by TR from the current study. It is worth pointing out that one might expect even higher levels of overlaps if not because of the following factors. First, genes that are indirectly regulated by T3 will not have TREs but can also be identified by RNA-Seq analysis. Second, some TRE-containing genes may be regulated by T3 only at specific time points transiently or much later, requiring some proteins to be made or degraded first. Since the RNA-Seq analysis was done for only one time point, it would miss such genes. Thus, the observed overlaps between the genes found by ChIP-Seq and RNA-Seq indicate that many genes regulated by T3 during the early stages of intestinal remodeling are, not surprisingly, directly TR target genes.

Our bioinformatics analyses of the TR-bound genes showed that for both wild-type and TRα$^{-/-}$ intestine, many tissue remodeling-related GO terms and biological pathways were, as expected, highly enriched among the genes bound by TR. Given the two of the major early events during intestinal metamorphosis are larval epithelial cell death and de novo formation of adult epithelial stem cells, which subsequently proliferate and differentiate to form the adult epithelium by the end of the metamorphosis[5,31], it is satisfactory to find that many GO terms and biological pathways related to apoptosis/cell death and stem cells were highly enriched among the genes bound by TR in both wild-type and TRα$^{-/-}$ intestine (note that TRα$^{-/-}$ tadpoles do complete metamorphosis, although with some delay in intestinal remodeling[11,18]).

Interestingly, many cell cycle and cell proliferation-related GO terms and pathways were highly enriched among genes bound by TR in premetamorphic intestine with and/or without 18 h T3 treatment. This is surprisingly since major cell proliferation, i.e., the massive proliferation of the adult epithelial cells, takes place following de novo formation of the stem cells and after larval cell death is largely complete[22]. During T3-induced metamorphosis, intestinal epithelial cell death peaks after 2 days of T3 treatment

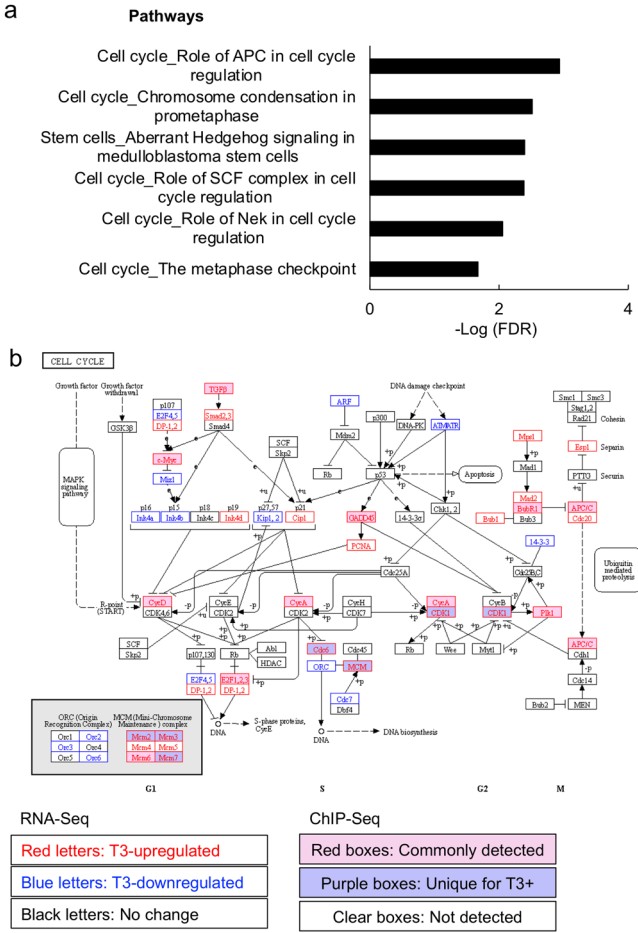

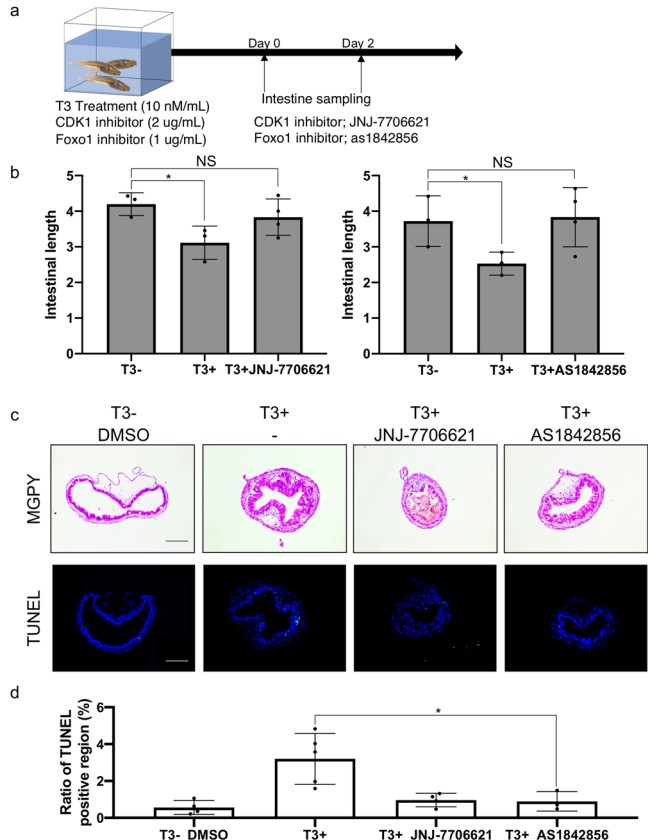

**Fig. 6 TRα-dependent enrichment of cell cycle pathways among genes bound by TR in the wild-type intestine. a** Top 10 cell cycle pathways differentially enriched among genes bound by TR in wild-type intestine. The pathways that were enriched among genes bound by TR in the wild-type intestine (Supplementary Data 6) were subtracted by the pathways that were enriched among genes bound by TR in TRα$^{-/-}$ intestine (Supplementary Data 12) to obtain the pathways only enriched in the wild-type intestine (Supplementary Data 14). The most significant 10 cell cycle-related pathways were plotted here (see Supplementary Data 20 for raw data). **b** TR-binding and regulation of genes in the pathway for cell cycle based on ChIP-Seq and RNA-Seq data. The pathway for cell cycle was visualized with regard to genes regulated by T3 based on RNA-Seq and/or bound by TR based on ChIP-Seq. The arrows show functional interaction: green for activation. Letters in red and blue indicate upregulated and downregulated genes, respectively, in wild-type intestine after T3 treatment. Letters in black indicate genes with no change in wild-type intestine after T3 treatment. Red boxes indicate ChIP-Seq detected genes both with and without T3 treatment, and purple boxes indicate genes identified by ChIP-Seq only in wild type with T3 treatment. Note that Cyclin A and CDK1 were among the genes directly bound by TR and upregulated by T3.

**Fig. 7 Inhibiting CDK1 signaling blocks intestinal remodeling during T3-induced metamorphosis. a** Experimental design for treatment with CDK1 and Foxo1 inhibitors. Wild-type tadpoles at stage 54 were treated with 2 μg/ml of CDK1 inhibitor JNJ-7706621 or 1 μg/ml of phosphorylation inhibitor of Foxo1 as1842856 for 2 days in the presence and absence of T3. Most of tadpoles treated with the inhibitors in the presence of T3 died at 3 days. Thus, intestinal remodeling was analyzed after 2 days of T3 treatment. **b** JNJ-7706621 and as1842856 inhibits T3-induced shortening of the intestine. Tadpoles at stage 54 were treated with or without 10 nM T3 for 2 days in the presence or absence of indicated inhibitor. Intestinal length was measured from bile duct junction to colon and normalized against body length. Each group included more than 4 tadpoles. Each bar represents the mean plus S.E. The asterisk (*) indicates a significant difference between the T3 treated tadpoles and control animals ($P < 0.05$) (see Supplementary Data 21 for raw data). **c** T3-induced intestinal morphological changes (epithelial folding) and apoptosis in the intestinal epithelium were inhibited by JNJ-7706621 and as1842856. Tadpoles were treated as above. Cross-sections of the intestine were stained with MGPY, which stains DNA blue and RNA red (top), and apoptotic cells were detected by TUNEL staining (bottom). TUNEL positive cells were stained green and nuclei were stained by Hoechst 33342 (blue). Bar indicates 20 μm (see Supplementary Data 21 for raw data). **d** Quantification of the apoptotic cells in **c**. TUNEL positive epithelium region was measured by using ImageJ software and normalized against the total epithelium region as determined by DAPI staining (see Supplementary Data 22 for raw data).

and significant cell proliferation occurs only after 3 days of T3 treatment[19]. These findings suggest that the activation of cell cycle and proliferation pathways is an early event important for larval cell death. Further support for this came from comparisons between the wild-type and TRα$^{-/-}$ intestine. First, there were many more cell cycle/proliferation-related GO terms and pathways enriched among TR-bound genes in the wild-type intestine (Supplementary Data 5, 6) than those enriched among TR-bound genes in the TRα$^{-/-}$ intestine (Supplementary Data 11, 12) (see

those present only in the wild-type intestine in Supplementary Data 13, 14). Second, TRα knockout dramatically inhibited T3-induced larval epithelial cell death and adult epithelial cell proliferation in the intestine[19], suggesting that TRα-dependent activation of the cell cycle/proliferation program is, surprisingly, important for larval epithelial cell death in addition to its likely involvement in subsequent adult epithelial stem cells proliferation.

Our studies with an inhibitor of CDK1, an important cell cycle protein, and an inhibitor of FOXO-1, a protein downstream of CDK1 and implicated to play a role in cell death[24,32] provided direct evidence to support a critical role of cell cycle activation in T3-induced larval epithelial cell death and the shortening of the intestine, which is likely due to larval epithelial cell death and intestinal contraction to ensure full coverage of the luminal surface of the intestine by the epithelium during intestinal remodeling. While our finding on a critical role of cell cycle activation in a large-scale cell death process is novel, especially with regard to vertebrate development, it is not unprecedented. One of the best-known transcription factor and cell cycle regulator is the oncogene c-Myc, which generally activates gene transcription to induce cell proliferation[33]. Interestingly, c-Myc overexpression in certain cell types, such as T-cells, can also induce cell death[34–43]. Thus, it is likely that over activation of cell cycle program in certain cell types may result in cell death. In this regard, it is worth noting that the larval epithelial cells in the tadpole intestine are differentiated yet mitotically active[5]. It is likely that T3 induces the cell cycle/proliferation pathways in the larval epithelial cells to facilitate their fate determination during intestinal remodeling: death via apoptosis or dedifferentiation to become adult stem cells. Thus, activation of the cell cycle/proliferation pathways in the larval epithelium may function as an important early step for both major changes, degeneration of the larval epithelium and formation of the adult epithelium, during T3-dependent intestinal metamorphosis.

T3 is not only important for amphibian metamorphosis but also critical for postembryonic development in mammals, a period around birth in human when many organs/tissues mature into their adult state. Identification and characterization of TR target genes in development is thus an important step toward understanding how T3 regulates vertebrate development. ChIP-Seq analyses have previously been carried out on cultured mouse cells and adult liver to identify many candidate TR target genes and corresponding biological pathways[44–46]. The lack of a properly assembled genome had made it difficult to perform ChIP-Seq analysis for genome-wide studies in frogs[47,48]. Our studies here represents the first genome-wide identification of TR target genes during vertebrate development, due in part to recent improvement in *Xenopus* genome sequence and annotation[49]. It is worth pointing out that most of the TR-binding peaks were in the intergenic regions but may have regulatory function through long range chromatin interactions. Unfortunately, they cannot be assigned to specific genes without chromosome interaction analysis. In any case, given the similarity and conservation between anuran metamorphosis and mammalian postembryonic development, it would be interesting to test in the future if similar GO terms and biological pathways are targeted by TR during mammalian development or in other *Xenopus* organs/tissues as we found here for the intestinal remodeling. Of particular interest is whether the regulation of cell cycle pathways is a common feature in the regulation of cell fate determination by T3 during vertebrate development and whether TRα and TRβ play similar roles in this.

## Methods
**Animals**. All *Xenopus tropicalis* experiments were approved by the Animal Use and Care Committee of Eunice Kennedy Shriver National Institute of Child Health and Human Development (NICHD), U.S. National Institutes of Health (NIH). Adult *Xenopus tropicalis* were purchased from NASCO (Fort Atkinson, WI, USA). Embryos were reared in 0.1 M Marc's modified Ringers (MMR) in petri dish for 1 day at 25 °C, and then transferred to an 800 mL beaker for 3 days. 4 days after fertilization, embryos were transferred into large volume (9-L) containers in a circulating system on a 12-h light/dark cycle. Tadpoles were staged according to the description for *X. laevis*[50].

**Chromatin immunoprecipitation (ChIP) sequencing (ChIP-Seq) and ChIP PCR**. Wild-type and knockout tadpoles at stage 54 were obtained as before[19]. Tadpoles

were treated with 10 nM T3 for 0 (control) or 18 h. The intestine was then isolated from these animals and flushed with 0.7× phosphate-buffered saline for chromatin isolation as before[51]. Intestines were placed in 1 ml of nuclei extraction buffer (0.5% Triton X-100, 10 mM Tris-HCl, pH 7.5, 3 mM CaCl2, 0.25 M sucrose, with the protease inhibitor tablet (Roche Applied Science, Complete, Mini, EDTA-free), 0.1 mM dithiothreitol in Dounce homogenizers on ice and crushed with 15–20 strokes by using pestle A (DWK Life Sciences (Kimble)). The homogenate was fixed in 1% formaldehyde with rotation at room temperature for 20 min, and the fixation was stopped with 0.1 M Tris-HCl, pH 9.5. The homogenate was centrifuged at $2000 \times g$ at 4 °C for 2 min, and the pellet was resuspended in 1 ml of nuclei extraction buffer and re-homogenized in Dounce homogenizers with 10–15 strokes using pestle B. Then the homogenate was filtered through a Falcon 70-μm cell strainer and centrifuged at $2000 \times g$ at 4 °C for 2 min. The pellet was resuspended in 200 μl of SDS lysis buffer (Merck Millipore Bioscience, MO, USA) on ice, sonicated by using the Bioruptor UCD-200 (Diagenode, Sparta). The output selector switch was set on High (H), and sonication was for 1 h. The samples were centrifuged at $16,000 \times g$ for 10 min at 4 °C. The chromatin in the supernatant was quantitated and frozen in aliquots at −80 °C. The DNA concentration of the chromatin was adjusted to 100 ng/μl by using the SDS lysis buffer, and then diluted to 10 ng/μl with ChIP dilution buffer (Merck Millipore Bioscience). After pre-clearing with salmon sperm DNA/protein A-agarose (Merck Millipore Bioscience), input samples were taken, and 500 μl of each chromatin sample was added to 1.5 ml tubes with anti-TRβ antibody and salmon sperm DNA/protein A-agarose beads and incubated with rotation for 4 h at 4 °C. After incubation, the beads were washed with 1 ml of ChIP Buffer I (0.1% SDS, 1% Triton X-100, 2 mM EDTA, 50 mM HEPES, pH 7.5, 150 mM NaCl), Buffer II (0.1% SDS, 1% Triton X-100, 2 mM EDTA, 50 mM HEPES, pH 7.5, 500 mM NaCl), Buffer III (0.25 M LiCl, 0.5% Nonidet P-40, 0.5% sodium deoxycholate, 1 mM EDTA, 10 mM Tris-HCl, pH 8.0), and TE (10 mM Tris-HCl, pH 8.1, 1 mM EDTA, pH 8.0) in succession. After the last wash, 100 μl of elution buffer (0.5% SDS, 0.1 M NaHCO3 (Sigma), 25 μg/ml proteinase K (Roche Applied Science) was added to the samples and input controls and the resulting mixture was rotated at 65 °C for 12 h. The ChIP DNA was purified by using the QIA-quick PCR purification kit (Qiagen) and eluted with 40 μl of EB buffer (Qiagen, 10 mM Tris-HCl, pH 8.5). Samples were analyzed by qPCR with a taqman probe against TRβ TRE region as describe before[48]. For high-throughput ChIP-Seq (Molecular Genomics Core, NICHD, NIH), libraries were prepared by DNA SMART™ ChIP-Seq kit (Clontech/Takara Bio Co., CA, USA) according to the manufacturer's protocol. The ChIP-Seq libraries were sequenced on the Illumina HiSeq 2500 platform, and three technical replicates were done for each sample. These raw read datasets for all ChIP-seq samples are available under Gene Expression Omnibus (GEO) accession number GSM4913228 to GSM4913239. ChIP-PCR was performed as described previously using primers as shown in the Supplementary Data 16 and 17[51].

**ChIP-Seq data processing**. Raw sequencing data in FASTQ format were aligned to *X. tropicalis* genome assemblies (Ensembl *Xenopus_tropicalis*.JGI_4.2 and Xenbase v9.1) with Bowtie2 software (version 2-2.3.4.1) and redundant reads were removed from final bam files with samtools software (version 1.9). Peak enrichments were detected with MACS2 software (version 2.1.1.20160309) by using the predicated fragment sizes of 52–122 bp set by default and q value cutoff of 0.05 for each single bam file without control, and a peak was assigned to a gene if it was mapped on to the coding region (including the introns) and/or 5 kb upstream or downstream of the cording region. Called peaks were mapped to genes in *Xenopus_tropicalis*.JGI_4.2.90.gff3 annotation with customer R scripts. This allowed us to identify TR-bound genes, i.e., genes with TR-binding peaks mapped on to the coding region (including the introns) and/or 5 kb upstream or downstream of the cording region. The false discovery rate (FDR) cutoff for peak calling was 0.05 for both ChIP-Seq and RNA-Seq analysis. To study the potential biological significance of the changes observed in the ChIP-Seq data, we performed pathway and gene ontology (GO) analyses with MetaCore software (GeneGo Inc., CA, USA), and used the KEGG Pathway database for visualization.

**Tadpole treatment with CDK inhibitor and foxo1 phosphorylation inhibitor**. Tadpoles at stage 54 were treated with either JNJ-7706621 (150 μg/150 mL water, AdooQ BioScience, CA, USA) or as1842856 (75 μg/150 mL water, AdooQ BioScience) during T3 induced metamorphosis. The water was changed every day, and T3 and inhibitors were added to each beaker. After 2 days of T3 treatment, tadpoles were euthanized and the intestine were fixed in 4% paraformaldehyde. Paraffin sections were stained with methyl green pyronin Y solution. Apoptotic cells in intestinal epithelium was detected by using Terminal deoxynucleotidyl transferase (TdT) dUTP Nick-End Labeling (TUNEL) assay with Click-iT ® Plus TUNEL Assay kit (Life Technologies, Inc., CA, USA).

**Statistical analysis**. Statistical analyses were performed by using the Prism 8 statistics software (GraphPad Software Inc., San Diego, CA) and Statcel computer software (OMS, Ltd., Saitama, Japan). Data were expressed as the mean ± SEM where applicable. P values were determined using Student's $t$-tests or Mann–Whitney $U$-tests. Differences with P values of less than 0.05 were considered significant.

**Reporting summary**. Further information on research design is available in the Nature Research Reporting Summary linked to this article.

## Data availability

The authors declare that the data supporting the findings of this study are available within the paper and Supplementary information and Supplementary data, and will be available from the corresponding author upon request. RNA-seq data were deposited at Gene Expression Omnibus (GEO), NCBI (GSM4913228 to GSM4913239).

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

## Acknowledgements

This work was supported by the intramural Research Program of NICHD, NIH. Y.T. and Y.S. were supported in part by Japan Society for the Promotion of Science (NIH) Fellowship. We thank Nga Luu for experimental help and laboratory management and the team in the Molecular Genomics Core, NICHD, NIH, for ChIP sequencing.

## Author contributions

Y.T. and Y.B.S. designed the research; Y.S. and Y.T. generated the knockout animals and extracted chromatin DNA samples; H.Z. assisted with the bioinformatics; All authors participated in the manuscript preparation and approve the final version of the manuscript.

## Funding

## Competing interests

The authors declare no competing interests.
