## [Transparent Peer Review File · Communications Biology]

Reviewers' comments:

Reviewer #1 (Remarks to the Author):

This manuscript describes a chromatin immunoprecipitation sequencing (ChIP-seq) experiment for thyroid hormone (T3) receptor conducted on the intestines of *Xenopus tropicalis* tadpoles treated with (T3) for 18 hours. Tadpole metamorphosis is a classical model for T3 action during development, and a genome-wide analysis of putative sites of TR association in chromatin is potentially a very valuable contribution to the field. The tadpole intestine undergoes extensive T3-dependent remodeling during metamorphosis, and Shi's lab has made major contributions to understanding the molecular developmental mechanisms underlying this process. In this paper they compared wild type and T3 receptor alpha (TRa) knockout animals that they had generated in the lab previously. They also treated tadpoles with a CDK1 or a FOXO1 inhibitor to investigate a potential role for cell cycle proteins in intestinal remodeling. They conclude that the TRa-dependent activation of cell cycle genes is responsible for cell death and adult epithelial stem cell development. This interpretation is similar to that which was recently published in the journal *Thyroid* where the authors used RNA sequencing. I have some questions about the data and the conclusions, and some suggestions that the authors may wish to consider.

The TR ChIP-seq experiment is potentially a very valuable dataset, but I think that the authors could do more with the analysis to make the results accessible to researchers in the field. For example, please explain more about how the experiment was conducted and whether any targeted validations were done. Also, it would be good if they could mine their data for more basic information as described below.

Lines 115-132, 156-190: I found the descriptions of the data (peak numbers and percentages) hard to follow. Below are some specific questions.

Lines 115-123: The authors indicate that they ensured "the reliability and reproducibility of the TR-bound genes..." in the three replicates, but it is unclear how well the replicates corresponded to each other. Looking at the data shown in the table in Fig. S2 there appears to have been a high degree of variability among replicates within each of the treatment groups. Was a principle components analysis done on the peak data? Also, in this paragraph the authors indicate that there were 11469 and 17930 peaks that corresponded to 3308 and 4319 TR-bound genes in WT intestine. Where were the other peaks located? Are they in intergenic regions? What p value was used to filter peaks? What window size was used to set peak width? How were peaks assigned to genes? Did the authors set a window upstream and downstream of the transcription start site (e.g., 5 or 10 kb) to call peaks associated with genes? Did they specifically look at peaks associated with differentially expressed genes? These total peak numbers (11469 and 17930) are different from what is given in the table in Fig. S2.

Please indicate what false discovery rate (FDR) cutoff was used for the ChIP-seq (and also the RNA-seq) analyses.

How many of the known T3-regulated genes in the intestine with characterized TREs had TR ChIP-seq peaks, and were these peaks in the predicted locations of the genome?

Can the authors load their datasets into the Integrated Genome Viewer to generate genome traces to make figures to illustrate their TR ChIP-seq data. If they also add their mapped RNA-seq data to the figure I think that this would enhance the presentation of the data.

I also think that it would strengthen the paper if some of the TR ChIP-seq peaks could be validated using targeted ChIP assays. It would be interesting to test if peaks identified by genome-wide analysis can be confirmed by a targeted approach.

Lines 124-132: Of the genes with TR ChIP-seq peaks (indicated as TR-bound genes), only 19% were induced and 9% were repressed by exogenous T3 (28% total). I think that this illustrates that there may be both technical and biological limitations to comparing the TR ChIP-seq and RNA-seq datasets, which can be discussed. But I don't think that the small percentage overlap between the two datasets necessarily validated your approach (line 132). Please explain.

Which known TRE motifs were found within the sequences covered by the peaks? How common was it to find the DR+4 sequence within the peaks? And how do the results compare to the few TR ChIP-seq experiments that have been conducted on mammalian cells? Are there other transcription factor binding sites that are found in proximity to TREs? These kinds of bioinformatics analyses are not that difficult to do, and I think that their inclusion in the paper would significantly enhance its importance.

Since the window used to assign peaks to genes is not indicated we don't know how far or close the peaks are to genes (and whether the genes called were differentially regulated by T3). Is the DR+4 sequence more commonly found associated with genes that are induced or repressed by T3? Are the TR ChIP-seq peaks more commonly found in upstream regions, within gene bodies or in intergenic regions? Do the TR ChIP-seq peaks found in intergenic regions have recognizable TREs?

It was unclear whether the GO analysis included all TR peaks, all peaks associated with genes, or just peaks associated with differentially expressed genes. I am unsure of the value of conducting GO analysis on genes with putative TR ChIP-seq peaks since such a large number of the peaks were assigned to genes that did not change expression. The RNA-seq GO analysis that the authors published in the journal *Thyroid* makes sense, since these are the genes that actually changed their expression level and so are involved in the biological processes under investigation. This may not be true for the TR ChIP-seq data. Only a fraction of the genes with TR ChIP-seq peaks changed expression, meaning that although they may have TR binding sites, they are not regulated by T3 (at least not in the tissue under investigation). However, a problem with this interpretation is that although a TR peak may be assigned to a gene based on proximity, this TR binding site may not regulate the gene. Conversely, TR binding sites without genes assigned may have regulatory function through long range chromatin interactions. The only way to tell is by chromosome interaction analysis. I think that it would be helpful to the reader to explain these caveats.

Lines 206-217. It would be good to list genes in the pathway that have TR binding sites and are regulated by T3 in the RNA-seq experiment; although I noticed that in the *Thyroid* paper, only 6 of 13 cell cycle genes identified by RNA-seq validated when analyzed by RTqPCR. How does this high false positive rate affect your bioinformatics analyses?

Although exogenous T3 can induce gene expression changes in premetamorphic tadpole tissues, some or perhaps many of these changes could be artifactual. I think that it is important to look at gene expression changes in the intestine during spontaneous metamorphosis. Have the authors analyzed any of the cell cycle genes identified in their RNA-seq experiment (and presumably confirmed as direct T3 response genes using ChIP-seq) throughout metamorphosis to confirm that their expression increases (or decreases) prior to or at the time of cell death? I worry that their result with exogenous T3 may not reflect what occurs during natural metamorphosis, and thus lead them to an erroneous conclusion.

Lines 221-222: The CDK1 inhibitor is well characterized in cell culture. Has it ever been used in living animals? The studies in whole tadpoles using the inhibitors are problematic since the compounds were clearly toxic to the animals. Tadpoles died after 3 days of treatment, so the authors sacrificed the animals after 2 days when they were probably quite sick. It might be informative to do the inhibitor experiments in organ culture.

Line 352: Did the controls receive the vehicle that T3 was dissolved in during the 18 hr that a group of tadpoles was treated with T3? The vehicle treatment is an important control.

Based on the GO analysis of genome-wide data and an experiment with two pharmacological inhibitors the authors concluded that the TR α -dependent activation of cell cycle genes is responsible for cell death and adult epithelial stem cell development. The authors recognize that this conclusion is “potentially surprising” (see line 220). I agree that it is surprising, and since it is counter to the prevailing paradigm in the field, I think that substantially more experimental evidence is required to draw this conclusion.

Minor editorial comments:

Line 49 – I think that you mean that there are two genes that code for two TR subtypes, not isoforms. Isoform is typically used to refer to differential processing of a protein (or pre-mRNA processing) to give different structural forms.

Line 55 – I don’t think the changes in mammalian organ structure from fetal to adult can necessarily be characterized as drastic. This adjective is appropriate for amphibians that undergo a metamorphosis, but not a direct developing species like a mammal. Please clarify what you mean.

Line 75 – It says that T3 was given for one day, but later it says that it was for 18 hr.

Lines 259-263: The increase in the number of TR ChIP-seq peaks after T3 treatment may represent increased TR recruitment to chromatin as the authors conclude. This could be due to T3 stabilizing the TR-RXR complex in chromatin, or increased expression of TR after T3 treatment, or both. Or it could be due to increased accessibility to antibodies of epitopes on TRs caused by T3 induction of chromatin remodeling. The result could be due to biology or a technical artifact, or perhaps both. The authors conclude that they detect low affinity TREs after T3 treatment. Please explain the basis for this hypothesis.

I could not see if the datasets have been submitted to GEO.

Reviewer #2 (Remarks to the Author):

The paper by Tanizaki and colleagues (YB Shi lab) describes a detailed molecular analysis of TR-binding on target genes in the remodelling intestine by ChIP-seq approach during T3-dependent metamorphosis. The study has been performed in T3-treated or untreated tadpoles in both WT and TR α -/- intestines. Moreover, comparison with previously acquired RNA-seq data by the same group allows to correlate mRNA expression and TR binding. Finally, one specific function targeted by T3-TR, cell cycle, has been approached more in depth and the authors clearly demonstrated the direct gene regulation of a cell cycle controlling gene.

The study is timely and gives novel and important information on direct T3-TR target genes in a developmental model. Importantly, the action of T3 is maintained in mammalian intestine. However, given the difficulty to perform a similar study in the mouse, the results are highly appreciated as they potentially suggest similar mechanisms in the developing gut in mammals.

I have only one minor comment for the authors. The Figure 3 (Hh pathway) and Figure 6B (chromosome condensation), but in particular Figure 3, are very difficult to follow. I strongly suggest to simplify the diagrams because there are too many colours and too many elements all together. Maybe by differentiating in separated panels activations and inhibitions the results would appear more clear and readable.

Responses to the reviewers' comments

We would like to thank the reviewers for the positive comments on our study and very helpful critics and suggestions. Below are the detailed changes and responses.

We hope that these changes and replies are satisfactory.

Reviewer #1:

This manuscript describes a chromatin immunoprecipitation sequencing (ChIP-seq) experiment for thyroid hormone (T3) receptor conducted on the intestines of *Xenopus tropicalis* tadpoles treated with (T3) for 18 hours. Tadpole metamorphosis is a classical model for T3 action during development, and a genome-wide analysis of putative sites of TR association in chromatin is potentially a very valuable contribution to the field. The tadpole intestine undergoes extensive T3-dependent remodeling during metamorphosis, and Shi's lab has made major contributions to understanding the molecular developmental mechanisms underlying this process. In this paper they compared wild type and T3 receptor alpha (TRa) knockout animals that they had generated in the lab previously. They also treated tadpoles with a CDK1 or a FOXO1 inhibitor to investigate a potential role for cell cycle proteins in intestinal remodeling. They conclude that the TRa-dependent activation of cell cycle genes is responsible for cell death and adult epithelial stem cell development. This interpretation is similar to that which was recently published in the journal *Thyroid* where the authors used RNA sequencing. I have some questions about the data and the conclusions, and some suggestions that the authors may wish to consider.

The TR ChIP-seq experiment is potentially a very valuable dataset, but I think that the authors could do more with the analysis to make the results accessible to researchers in the field. For example, please explain more about how the experiment was conducted and whether any targeted validations were done. Also, it would be good if they could mine their data for more basic information as described below.

Answer: We appreciate the suggestion. We have now added such explanations.

Lines 115-132, 156-190: I found the descriptions of the data (peak numbers and percentages) hard to follow. Below are some specific questions. Lines 115-123: The authors indicate that they ensured "the reliability and reproducibility of the TR-bound genes..." in the three replicates, but it is unclear how well the replicates corresponded to each other. Looking at the data shown in the table in Fig. S2 there appears to have been a high degree of variability among replicates within each of the treatment groups. Was a principle components analysis done on the peak data?

Answer: We apologize for not explaining it carefully. We have revised the text. As there would be expectedly some variations among the replicates that might produce false positive peaks, we reasoned that the peaks that were present in all three replicates would be much more likely to be true TR binding sites. Thus, we focused on such peaks. We did not do PCA on the raw peak data. The variations among replicates in Fig. S2 were not unexpected as individual replicates were

treated separately and likely might have different PCR amplification efficiencies to produce different levels of total reads. Unlike RNA-Seq, where raw data can be converted to FPKM (fragments per kilobase of exon model per million mapped reads), there is no easy way to normalize the raw ChIP-seq data. That was why we focus on individual peak locations, not peak value. We have also revised the text to make it clearer.

Also, in this paragraph the authors indicate that there were 11469 and 17930 peaks that corresponded to 3308 and 4319 TR-bound genes in WT intestine. Where were the other peaks located? Are they in intergenic regions?

Answer: TR-binding peaks could locate upstream, downstream, or within a gene. Thus, on the average, there were several peaks per each TR-bound gene. In addition, there were some binding sites far from transcribed regions and thus could not be assigned to any genes based on the program used in our analysis. These peaks are indicated as Not assigned (NA) in Figs. S1-2. We have added the detailed description in Figure 1A legend to clarify this.

What p value was used to filter peaks?

Answer: We have described this in the material and method, the peaks were called if q value is ≤ 0.05 .

What window size was used to set peak width? How were peaks assigned to genes?

Answer: We used default value in MACS2 program for peak calling. A peak was assigned to a gene if it was within 5 kb upstream of the start site or in the gene body, including introns. We have added the details in the methods.

Did the authors set a window upstream and downstream of the transcription start site (e.g., 5 or 10 kb) to call peaks associated with genes?

Answer: The peaks were called first and then assigned to genes with a window of 5 kb upstream of the start site or within the gene body (see above) (Table S4).

Did they specifically look at peaks associated with differentially expressed genes?

Answer: No, we did not. We simply used ChIP-seq data to identify TR-bound genes and then compared to the differentially expressed genes identified previously. We thought that this was the more appropriate and less biased way.

These total peak numbers (11469 and 17930) are different from what is given in the table in Fig. S2.

Answer: In the original Figure S2, all peaks found in the individual replicates were included. On the other hand, the peaks in Fig. 1 and the supplement tables had only peaks present in all three replicates. We have now added the information in the revised Fig. S2. In addition, we have added the explanation in the manuscript (Result section).

Please indicate what false discovery rate (FDR) cutoff was used for the ChIP-seq (and also the RNA-seq) analyses.

Answer: FDR cutoff for peak calling was 0.05 for both ChIP-Seq and RNAseq analysis. We have added the information in the materials and methods.

How many of the known T3-regulated genes in the intestine with characterized TREs had TR ChIP-seq peaks, and were these peaks in the predicted locations of the genome?

Answer: There were only a limited number of T3-regulated genes with characterized TREs in *Xenopus tropicalis*. Two of the best known are TR β and TH/bZip. We analyzed these two genes and indeed found that TRE regions of these genes were bound by TR based on ChIP-Seq in this study. We have added the information in Figure S4 and Results.

Can the authors load their datasets into the Integrated Genome Viewer to generate genome traces to make figures to illustrate their TR ChIP-seq data. If they also add their mapped RNA-seq data to the figure I think that this would enhance the presentation of the data.

Answer: Yes, we have now added two examples in Fig. S4.

I also think that it would strengthen the paper if some of the TR ChIP-seq peaks could be validated using targeted ChIP assays. It would be interesting to test if peaks identified by genome-wide analysis can be confirmed by a targeted approach.

Answer: We have randomly selected a few genes identified by ChIP-Seq based on high values of fold enrichment and distance from the coding region and carried out ChIP-PCR analysis to confirm the TR binding to the TRE region. The data have been added in the new Fig. S5.

Lines 124-132: Of the genes with TR ChIP-seq peaks (indicated as TR-bound genes), only 19% were induced and 9% were repressed by exogenous T3 (28% total). I think that this illustrates that there may be both technical and biological limitations to comparing the TR ChIP-seq and RNA-seq datasets, which can be discussed. But I don't think that the small percentage overlap between the two datasets necessarily validated your approach (line 132). Please explain.

Answer: The overlaps were expected to be relatively low since not all genes bound by TR would be regulated within the T3 treatment time or at only a single time point as some might require other proteins to be made first or only transiently regulated by T3. Conversely, some of the regulated genes by T3 treatment might be indirectly regulated. We have not added the explanation to the Results and Discussion.

Which known TRE motifs were found within the sequences covered by the peaks?

Answer: Studies in frog TR target genes are limited and only DR+4 TREs have been reported. We did find such TREs were highly enriched in the TR-binding peaks (Fig. S3). Two examples are shown in new Fig. S4.

How common was it to find the DR+4 sequence within the peaks?

And how do the results compare to the few TR ChIP-seq experiments that have been conducted on mammalian cells?

Answer: We carried out motif analysis for all peaks and found that DR+4 sequences were highly enriched in the TR-binding peaks (Fig. S3), similar to that observed in mammals.

Are there other transcription factor binding sites that are found in proximity to TREs? These kinds of bioinformatics analyses are not that difficult to do, and I think that their inclusion in the paper would significantly enhance its importance.

Answer: We analyzed other transcription factor binding sites in proximity to TREs, but could not make any significant conclusions. Thus, we have not included such information.

Since the window used to assign peaks to genes is not indicated we don't know how far or close the peaks are to genes (and whether the genes called were differentially regulated by T3).

Answer: As indicated above, we used a 5 kb window for upstream peaks. All peak positions for TR-bound genes are shown in Table S1-S2 and S7-S8. We have also discussed the overlaps between TR-bound genes and T3-regulated genes as indicated above.

Is the DR+4 sequence more commonly found associated with genes that are induced or repressed by T3?

Answer: Indeed, we found that TR-bound genes with DR4 sequences were enriched among genes induced by T3.

Are the TR ChIP-seq peaks more commonly found in upstream regions, within gene bodies or in intergenic regions?

Answer: Most of peaks were intergenic as shown in Table S4.

Do the TR ChIP-seq peaks found in intergenic regions have recognizable TREs?

Answer: The only known recognizable TREs are the DR+4 like sequences, which were enriched.

It was unclear whether the GO analysis included all TR peaks, all peaks associated with genes, or just peaks associated with differentially expressed genes. I am unsure of the value of conducting GO analysis on genes with putative TR ChIP-seq peaks since such a large number of the peaks were assigned to genes that did not change expression. The RNA-seq GO analysis that the authors published in the journal Thyroid makes sense, since these are the genes that actually changed their expression level and so are involved in the biological processes under investigation. This may not be true for the TR ChIP-seq data. Only a fraction of the genes with TR ChIP-seq peaks changed expression, meaning that although they may have TR binding sites,

they are not regulated by T3 (at least not in the tissue under investigation). However, a problem with this interpretation is that although a TR peak may be assigned to a gene based on proximity, this TR binding site may not regulate the gene.

Answer: We appreciate the comments. Indeed, this is a complicate issue. All GO and pathway analyzes are based on the genes with TR-ChIP peaks, not on differentially expressed genes as in our earlier study. It is true that only a fraction of the bound genes were found to be regulated based on our earlier RNA-Seq data. However, that RNA-Seq data were limited to one T3 treatment time point. It is possible that longer or shorter treatment time may affect many other TR-bound genes. Thus, we feel that our analyses are valuable as they reveal potential regulation by TR directly. In fact, we found the cell cycle pathway was significantly affected directly by TR based on our current analysis. Although the earlier RNA-seq analysis also came to the same conclusion, many of the T3 regulated genes by RNA-seq might be indirect T3-regulated genes. It is our analysis here that demonstrates that cell cycle regulation is one of the earliest events affected by T3.

Conversely, TR binding sites without genes assigned may have regulatory function through long range chromatin interactions. The only way to tell is by chromosome interaction analysis. I think that it would be helpful to the reader to explain these caveats.

Answer: We agree with the reviewer. We have pointed out this in the discussion.

Lines 206-217. It would be good to list genes in the pathway that have TR binding sites and are regulated by T3 in the RNA-seq experiment; although I noticed that in the Thyroid paper, only 6 of 13 cell cycle genes identified by RNA-seq validated when analyzed by RTqPCR. How does this high false positive rate affect your bioinformatics analyses?

Answer: Thanks for the suggestion. In the Thyroid paper, the 6 out of 13 genes were those regulated in the wild type and also affected by TRa KO. In fact, with a few exceptions, the 13 genes were indeed regulated by T3 by 2 fold or more in the WT as shown in the Fig. 5 of our earlier publication. We have now listed the genes in the pathway that have both TR-binding and T3-regulation in Supplemental Table S15.

Although exogenous T3 can induce gene expression changes in premetamorphic tadpole tissues, some or perhaps many of these changes could be artifactual. I think that it is important to look at gene expression changes in the intestine during spontaneous metamorphosis. Have the authors analyzed any of the cell cycle genes identified in their RNA-seq experiment (and presumably confirmed as direct T3 response genes using ChIP-seq) throughout metamorphosis to confirm that their expression increases (or decreases) prior to or at the time of cell death? I worry that their result with exogenous T3 may not reflect what occurs during natural metamorphosis, and thus lead them to an erroneous conclusion.

Comparison between nf54 and 61 is one of the methods as reviewer suggested, but it is hard to evaluate “T3 target”. Also T3 stimulates apoptosis of epithelial cells and stem cell proliferation,

and this event is the same to natural metamorphosis in intestine and many papers have been reported to support this fact.

Lines 221-222: The CDK1 inhibitor is well characterized in cell culture. Has it ever been used in living animals? The studies in whole tadpoles using the inhibitors are problematic since the compounds were clearly toxic to the animals. Tadpoles died after 3 days of treatment, so the authors sacrificed the animals after 2 days when they were probably quite sick. It might be informative to do the inhibitor experiments in organ culture.

Answer: The reviewer correctly pointed out the potential problems with T3 treatment. We fully appreciate this. On the other hand, we and others have published many studies on gene expression in the intestine during metamorphosis or by T3 treatment, the results have always been consistent. It is likely that we may find some discrepancies for genes with lower folds of regulation. It is difficult for us to analyze many genes. Regarding the inhibitor studies, unfortunately, organ cultures are not suitable for analyses of tissue morphology or intestinal length reduction with or without T3 treatment.

Line 352: Did the controls receive the vehicle that T3 was dissolved in during the 18 hr that a group of tadpoles was treated with T3? The vehicle treatment is an important control.

Answer: The stock solution of T3 was in NaOH (10 mM/L) and added to the rearing water after about 10,000 fold dilution without affecting the PH of the rearing water. Thus, the vehicle was essentially adding water. In our many years of experience with tadpoles, adding such minute quantity of NaOH to the rearing water does not affect the tadpoles.

Based on the GO analysis of genome-wide data and an experiment with two pharmacological inhibitors the authors concluded that the TRa-dependent activation of cell cycle genes is responsible for cell death and adult epithelial stem cell development. The authors recognize that this conclusion is “potentially surprising” (see line 220). I agree that it is surprising, and since it is counter to the prevailing paradigm in the field, I think that substantially more experimental evidence is required to draw this conclusion.

Answer: We do agree that more experimental evidence would better support our conclusion. Unfortunately, other than pharmacological inhibition, the only possibility is to use conditional knockout, which is impossible do in *Xenopus* at the present time. Whole animal knockout or of cell cycle genes would cause lethality in development. Thus, we hope that the reviewer would agree that such studies are beyond the scope of this study. We should also point out that as we discussed, it was also surprising when the oncogene c-Myc was first found to induce apoptosis. Thus, it is possible that cell cycle can be activated in some cell death processes as well. Our findings here may help to raise interest for future studies on this surprising phenomenon.

Minor editorial comments:

Line 49 – I think that you mean that there are two genes that code for two TR subtypes, not isoforms. Isoform is typically used to refer to differential processing of a protein (or pre-mRNA

processing) to give different structural forms.

Answer: We have changed isoform to subtype.

Line 55 – I don't think the changes in mammalian organ structure from fetal to adult can necessarily be characterized as drastic. This adjective is appropriate for amphibians that undergo a metamorphosis, but not a direct developing species like a mammal. Please clarify what you mean.

Answer: Some organs do change quite drastically as they mature. In particular, intestine had no crypt at birth in mouse and the crypt is formed during maturation after birth. In any case, we removed "drastic" from introduction.

Line 75 – It says that T3 was given for one day, but later it says that it was for 18 hr.

Answer: We have made the correction throughout manuscript.

Lines 259-263: The increase in the number of TR ChIP-seq peaks after T3 treatment may represent increased TR recruitment to chromatin as the authors conclude. This could be due to T3 stabilizing the TR-RXR complex in chromatin, or increased expression of TR after T3 treatment, or both. Or it could be due to increased accessibility to antibodies of epitopes on TRs caused by T3 induction of chromatin remodeling. The result could be due to biology or a technical artifact, or perhaps both. The authors conclude that they detect low affinity TREs after T3 treatment. Please explain the basis for this hypothesis.

Answer: We agree with the reviewer's comments and have added revised the text.

I could not see if the datasets have been submitted to GEO.

Answer: These raw read datasets are available for all ChIP-seq samples under Gene Expression Omnibus (GEO) accession number GSM4913228 to GSM4913239. We added the information in the materials and methods.

Reviewer #2

The paper by Tanizaki and colleagues (YB Shi lab) describes a detailed molecular analysis of TR-binding on target genes in the remodelling intestine by ChIP-seq approach during T3-dependent metamorphosis. The study has been performed in T3-treated or untreated tadpoles in both WT and TRa^{-/-} intestines. Moreover, comparison with previously acquired RNA-seq data by the same group allows to correlate mRNA expression and TR binding. Finally, one specific function targeted by T3-TR, cell cycle, has been approached more in depth and the authors clearly demonstrated the direct gene regulation of a cell cycle controlling gene.

The study is timely and gives novel and important information on direct T3-TR target genes in a developmental model. Importantly, the action of T3 is maintained in mammalian intestine. However, given the difficulty to perform a similar study in the mouse, the results are highly appreciated as they potentially suggest similar mechanisms in the developing gut in mammals.

I have only one minor comment for the authors. The Figure 3 (Hh pathway) and Figure 6B (chromosome condensation), but in particular Figure 3, are very difficult to follow. I strongly suggest to simplify the diagrams because there are too many colours and too many elements all together. Maybe by differentiating in separated panels activations and inhibitions the results would appear more clear and readable.

Answer: We appreciate the comments. We have changed the pathway from the map derived from Metacore software to a new map based on the David software. We hope that the new ones are clearer.

REVIEWERS' COMMENTS:

Reviewer #1 (Remarks to the Author):

The authors have responded thoughtfully to my questions and concerns. I have no further comments.

Reviewer #2 (Remarks to the Author):

The authors have considered my comments and modified their paper.